

# Predicting Chinese stock market using XGBoost multi-objective optimization with optimal weighting

Jichen Liu

School of International Trade and Economics, University of International Business and Economics, Beijing, China

## ABSTRACT

The application of artificial intelligence (AI) technology in various fields has been a recent research hotspot. As a representative technology of AI, the specific application of machine learning models in the field of economics and finance undoubtedly holds significant research value. This article proposes Extreme Gradient Boosting Multi-Objective Optimization Model with Optimal Weights (OW-XGBoost) to comprehensively balance the returns and risks of investment portfolios. The model utilizes fusing label with optimal weights to achieve multi-objective tasks, effectively controlling the impact of various risk and return indicators on the model, thus improving the interpretability and generalization ability of the model. In the experiments, we tested the model using China A-share data from October 2022 to April 2023 and conducted a series of robustness tests. The results indicate that: (1) The OW-XGBoost outperforms the XGBoost Model with Yield as Label (YL-XGBoost), XGBoost Multi-Label Classification Model (MLC-XGBoost) in controlling risk or achieving returns. (2) OW-XGBoost performs better overall compared to baseline models. (3) The robustness tests demonstrate that the model performs well under different market conditions, stock pools, and training set durations. The model performs best in moderately fluctuating stock markets, stock pools comprising high market value stocks, and training set durations measured in months. The methodology and results of this study provide a new perspective and approach for fundamental quantitative investment and also create new possibilities and avenues for the integration of AI, machine learning, and financial quantitative research.

## INTRODUCTION

Recently, there has been a substantial increase in the incorporation of artificial intelligence (AI) technologies in the fields of economics and finance (*Xu et al., 2022*; *Liu et al., 2024*). Significantly, prominent institutions such as Citibank, JPMorgan, and Two Sigma have utilized artificial intelligence (AI) for a wide range of objectives. These include developing strong anti-fraud systems, forecasting monetary policies, and identifying investment opportunities by conducting thorough analysis of financial reports and news content (*Li & Sun, 2019*; *Leow, Nguyen & Chua, 2021*). These practical examples vividly demonstrate the

Corresponding author
Jichen Liu,
liujichen202109@163.com

numerous potential applications of AI technology in the financial sector (*Li & Sun, 2020*; *Luo, Zhuo & Xu, 2023*).

The significant progress and profound effects of AI have positioned it as a focal point in global competition, leading nations to prioritize it as a crucial area that warrants significant government backing. In this context, from January 2020 to June 2023, several ministries and commissions of the Chinese central government released a total of 17 policies and guidance documents that specifically targeted different industries. Significant attention should be given to the joint endeavor of six government departments, namely the Ministry of Science and Technology and the Ministry of Industry and Information Technology. This effort resulted in the publication of "Guiding Opinions on Accelerating Scenario Innovation to Foster High-quality Economic Development through Advanced AI Applications" in June 2022. This directive highlighted the importance of investigating the utilization of AI technology in key sectors, including manufacturing, agriculture, logistics, finance, business, and home improvement (*China Ministry of Science and Technology, 2022*).

Considering the significant importance of this government guidance, it is crucial to thoroughly examine the various situations in which machine learning, a fundamental aspect of AI technology, plays a crucial role (*Taddy, 2018*). These scenarios represent a wide range of possible uses, requiring thorough investigation and analysis.

Machine learning has been widely utilized in the fields of economics and finance, encompassing a range of tasks such as data analysis (*Glaeser et al., 2016*; *Jean et al., 2016*), text processing (*Hansen, McMahon & Prat, 2017*; *Larsen, 2021*), and economic forecasting (*Mullainathan & Spiess, 2017*; *Chalfin et al., 2016*). Significantly, well-known quantitative investment firms such as Renaissance Technologies' Medallion Fund, Two Sigma, Applied Quantitative Research, and Bridgewater Associates have made substantial investments in utilizing machine learning technologies for selecting stocks, timing trades, and implementing various quantitative investment strategies. The combined endeavors have resulted in significant profits, highlighting the effectiveness of machine learning applications in enhancing their financial pursuits.

There is a significant amount of scholarly literature that explores portfolio investment, which has evolved from Markowitz's groundbreaking mean-variance model. The investment theory in modern finance has continuously progressed since that time. *Sharpe (1964)* and *Ross (1976)* made significant contributions to the field by introducing the capital asset pricing model and the arbitrage pricing theory, which played a crucial role in advancing this area of study. In addition to the arbitrage pricing theory, *Fama & French (1993)* developed the three-factor model and later the five-factor model (*Fama & French, 2015*), progressively establishing a framework to incorporate complex models in this field.

With the advancement of AI, an increasing number of researchers are now investigating the use of machine learning techniques in selecting factors for multi-factor models (*Zou & Zhang, 2009*; *Xu & Ghosh, 2015*), predicting stock prices or indices (*Gu, Kelly & Xiu, 2020*; *Freitas, De Souza & de Almeida, 2009*; *Zhang, Chu & Shen, 2021*; *Chen & Hao, 2017*; *Tsai & Wang, 2009*; *Huang, Yang & Chuang, 2008*), and exploring other research domains. In

the realm of quantitative investment research utilizing machine learning, the primary accomplishments thus far can be categorized into three general areas as outlined below.

Firstly, research has investigated the effectiveness of various machine learning models in stock investment. For example, *Leippold, Wang & Zhou (2022)* examined the effectiveness of different machine learning models in forecasting complex indicators. This study also examined the importance of various indicator types within these models, the influence of retail investors on the dynamics of the stock market, and differences among stocks based on market capitalization and corporate characteristics. *Vrontos, Galakis & Vrontos (2021)* performed a comparative analysis between machine learning models and statistical econometric models. Their assessment, which considered economic and statistical perspectives, emphasized the superior ability of machine learning models to forecast implied volatility. *Huang, Yang & Chuang (2008)* conducted a comparative evaluation of various machine learning models, such as Wrapper, support vector machine (SVM), K-nearest neighbors (KNN), backpropagation neural network (BP neural network), decision trees, and logistic regression. The study utilized stock market data from Taiwan and Korea and found that the Wrapper method performed better in predicting these markets. Nevertheless, they also highlighted the praiseworthy precision in forecasting accomplished by alternative machine learning models. In addition, *Gu, Kelly & Xiu (2020)* utilized prominent machine learning models to systematically forecast the trends of the US market. They then compared these results with those obtained from linear regression models. Their thorough analysis offered perceptive viewpoints on the efficacy of various models in forecasting stock market behavior.

Secondly, there have been notable improvements in enhancing well-established machine learning algorithms. *Chen & Hao (2017)* enhanced the traditional support vector machine model by incorporating an information gain feature weighting approach. Integrating this innovation with KNN resulted in significant enhancements in predictive performance, particularly in the field of stock index prediction. In a similar manner, *Freitas, De Souza & de Almeida (2009)* presented a self-regressive moving reference neural network and demonstrated its outstanding performance by creating an investment portfolio model. Their study showcased the superiority of this approach in modeling investment portfolios, emphasizing its effectiveness in real-world financial applications. In addition, *Zhang, Chu & Shen (2021)* introduced the SVR-ENANFIS model, which offers an improved iteration compared to the conventional ENANFIS model. Their research demonstrated that the SVR-ENANFIS model exhibited superior accuracy in forecasting stock prices, showcasing significant advancements compared to current methodologies.

Thirdly, the assessment of combined models in the stock market domain involves the fusion of different machine learning models or the integration of machine learning models with their statistical counterparts. *Tsai & Wang (2009)* utilized decision trees and artificial neural networks in a customized ensemble algorithm designed for the purpose of predicting stock prices. Based on data from the Taiwanese stock market, their research determined that the ensemble algorithm outperformed the individual algorithms in terms of predictive accuracy, highlighting the superiority of this combined methodology. Furthermore, there have been significant improvements in enhancing well-established

machine learning algorithms. *Chen & Hao (2017)* enhanced the traditional support vector machine model by incorporating an information gain feature weighting approach. Integrating this innovation with KNN resulted in significant improvements in predictive performance for stock index prediction. Similarly, *Freitas, De Souza & de Almeida (2009)* presented a self-regressive moving reference neural network and demonstrated its outstanding performance by creating an investment portfolio model. Their study showcased the dominance of this approach in simulating investment portfolios, emphasizing its effectiveness in real-world financial scenarios. In addition, *Zhang, Chu & Shen (2021)* introduced the SVR-ENANFIS model, which offers an improved iteration compared to the conventional ENANFIS model. Their research demonstrated that the SVR-ENANFIS model exhibited superior accuracy in forecasting stock prices, showcasing significant advancements compared to current methodologies.

Although there has been progress in various areas, current research on this subject often demonstrates several constraints. First, the construction of the label seems to be quite simple. The majority of studies depend on returns, stock prices, or labels derived from returns and stock prices (*Gu, Kelly & Xiu, 2020*; *Zhang, Chu & Shen, 2021*; *Tsai & Wang, 2009*). Second, the importance of these labels is typically one-dimensional, mainly dependent on returns. This statement lacks comprehensive inclusion of vital risk information that is indispensable in practical investment scenarios, resulting in difficulties in achieving a harmonious equilibrium between returns and risks. Third, many studies lack thorough validation of their results. Backtesting primarily involves employing a singular methodology, utilizing either a single stock pool or a restricted set of targets for testing. Furthermore, the backtesting period is typically limited, assessing the efficacy of the model within distinct market phases, such as periods of market optimism or stable stock market conditions.

This study aims to develop a methodology for creating composite labels that combine multiple indicators. The purpose of these labels is to provide a thorough representation of the risks and rewards associated with stocks. These composite labels will be used as the basis for training machine learning models and assessing their effectiveness in market dynamics. The main contributions of this research are outlined as follows:

- The fusion labels have been carefully created by combining information from five dimensions: returns, excess returns, volatility, maximum drawdown, and Sharpe ratio. This integration allows the labels to comprehensively encompass the various aspects of risk and return that are relevant to investment targets. Furthermore, the grid search method has been employed to determine the optimal weights for constructing these fusion labels.

- This study examines the market performance of OW-XGBoost and compares its efficacy to that of YL-XGBoost, MLC-XGBoost, and other notable machine learning models.

- An extensive analysis of the robustness of OW-XGBoost has been performed, considering different stock pools, diverse market conditions, and varying lengths of training set periods. The objective of this investigation is to assess the model's reliability and flexibility in the face of changing market conditions and varying data accessibility.

# MODEL DEVELOPMENT

Until now, most studies have exclusively used individual return indicators, such as rate of return or stock price, as the main labels in their research. Nevertheless, this approach has limitations due to two primary factors:

- Disregarding market volatility weakens the ability of models to make accurate predictions. Volatility has a substantial effect on the fluctuations in stock prices, affecting both the extent and the rate at which they change. Neglecting this aspect leads to prediction models that are unable to accurately forecast future price movements, thus withholding important insights into potential trends and price ranges. Furthermore, market volatility is crucial in differentiating between random fluctuations in prices and significant patterns. Not considering volatility can result in mistaking random fluctuations for long-term trends, which can lead to inaccurate predictions.

- Single-return indicators fail to account for the intricate and volatile characteristics of the market. Anticipating high-quality investment opportunities involves navigating complex, nonlinear connections among multiple influential factors. Single indicators face difficulties in fully encompassing this complex reality. Moreover, the inherent volatility of stock market data makes models that rely on single-return indicators excessively responsive to data noise or anomalies, thereby compromising the accuracy of predictions.

In order to overcome these constraints, we developed a multi-objective optimization model using XGBoost, a highly acclaimed algorithm known for its outstanding performance and efficiency. XGBoost, in contrast to conventional machine learning models, provides resilience and the ability to manage overfitting by utilizing parameters such as pruning. While prior studies primarily utilized XGBoost with rate of return or stock prices as labels, our objective is to introduce innovation by utilizing fused labels for training. The empirical findings in the subsequent sections of this study clearly show that this model has superior predictive capabilities.

## Selection of risk and return indicators

In order to provide a comprehensive depiction of both returns and risk, our study has chosen five crucial indicators. When it comes to returns, we take into account both the rate of return and the excess return. The rate of return represents the actual increase in assets, whereas the excess return indicates the proficiency in generating additional returns through a particular investment strategy. Regarding risk assessment, our attention is directed towards the metrics of standard deviation and maximal drawdown. The standard deviation measures the consistency of returns generated by the investment strategy, while the maximal drawdown indicates the asymmetry and peakedness in the distribution of returns, addressing specific shortcomings of the standard deviation. Moreover, the Sharpe ratio quantifies the additional return gained in relation to the assumed level of risk, which is of utmost importance for investors when making decisions.

The chosen indicators create a composite that better captures the various aspects of stock returns and risks compared to relying solely on return rates or stock prices. This composite is crucial as it possesses qualities that are more suitable for machine learning models. It allows for the identification of complex and nonlinear relationships among features in the market, which is both intricate and unstable. The calculation methods for the five indicators are as follows.

Let $T_i$ symbolize the net asset value (NAV) on the $i^{th}$ day, while $R_p$ signifies the cumulative return of the investment portfolio. $R_f$ stands for the risk-free rate, and $\beta_p$ denotes the beta value of the portfolio. We have:

Cumulative return:

$$R_p = \frac{T_i - T_0}{T_0}. \tag{1}$$

Excess return:

$$Alpha = R_p - \left( R_f - \beta_p \times (R_m - R_f) \right). \tag{2}$$

Standard deviation:

$$\sigma_p = \sqrt{\frac{1}{N} \sum_{i=1}^{N} (R_i - \overline{R_p})^2}. \tag{3}$$

Max Drawdown:

$$MaxDrawdown = \frac{\max(T_m - T_n)}{T_m}, \tag{4}$$

where $T_i$ represents the net value of the composition on day $i$.

SharpeRatio:

$$SharpeRatio = \frac{R_p - R_f}{\sigma_p}. \tag{5}$$

The process employed to generate labels for the accumulated return rate and excess return rate entails assigning a value of 1 to samples that are greater than 0, while assigning a value of 0 to all other samples. Nevertheless, when it comes to standard deviation, maximal drawdown, and Sharpe ratio, samples that are lower than the sample mean are designated with a label of 1, while all other samples are assigned a label of 0.

## Fusing labels based on optimal weights

The use of multi-objective optimization methods demands the simultaneous realization of profit acquisition and risk control objectives. In this way, they closely resemble the multi-label classification problems seen in the machine learning field. Researchers have proposed various approaches to address such issues, which can broadly be divided into traditional multi-label classification methods (*Moyano et al., 2018*; *Zhang & Zhou, 2014*) and deep

learning-based multi-label classification methods (*Kim, 2014*; *Vaswani et al., 2017*). Yet, the aforementioned methods are subject to certain limitations when applied in the financial domain, such as the difficulties encountered when addressing label correlation and label imbalance issues, as well as the potential for low model interpretability and weak generalization capabilities.

Hence, this article introduces the concept of "fusing labels". The construction process is as follows: (1) assign specific weights to each label; (2) sum the products of multiple labels and their weights, with the resulting sum taken as the fused label value. Using this approach, multi-labels are transformed into a single composite label that merges both return and risk information, hence the term "fusing labels". The detailed fusion process is as follows:

To ensure optimal model performance, the best weights are filtered out for label construction. These weights possess the following characteristics: (1) the weight of a single indicator falls between 0 and 1; (2) the value of the fusing labels obtained based on the weight maps falls between 0 and 1; and (3) the minimum change unit for each weight is 0.1.

The fusing labels method based on optimal weights offers the following advantages:

1) Adjustable weights: The impact of various indicators on the model can be influenced by adjusting the weights. This allows for the theoretically effective analysis of the individual contributions of each indicator and practical adjustments to the model to be made based on economic finance theory, historical experience, or investor risk-return preferences.

2) Alleviating label correlation issues: Fusing labels can eliminate redundant information introduced by correlations, thereby enhancing the model's prediction accuracy.

3) Mitigating label imbalance problems: Multi-label scenarios can easily produce label imbalances. As a result, compared to single-label models, the trained model will be biased towards high-frequency labels or label combinations. Fusing labels can alleviate this, enabling the model to equally address different categories of samples and improve its prediction capabilities in relation to minority classes.

4) Enhancing model interpretability: Financial applications entail the use of models with high interpretability, necessitating an understanding and explanation of the model's prediction results. By converting multi-label classification problems into single-label classifications, fusing labels can offer a better, more comprehensive understanding of the model's predictive outcomes.

5) Resolving label conflict issues: When dealing with financial data, different labels might inherently conflict with each other, such as return labels and risk labels, making it more difficult for the model to understand the labels. The use of fusing labels can avoid such conflicts.

6) Improving generalization capabilities: Multi-label classification generally produces more complex models with a higher probability of overfitting. The utilization of fusing labels simplifies the model structure and suppresses the overfitting issue, enhancing the model's generalization capabilities in the process.

## OW-XGBoost

The XGBoost algorithm (*Chen & Guestrin, 2016*) is an optimized algorithm based on boosting tree algorithms, such as Adaptive Boosting (Adaboost) and Gradient Boosting Decision Tree (GBDT). Specifically, it learns by integrating multiple weak classifiers. Given a training set $D = (x_i, y_i)$, where $D$ contains $n$ records and $m$ variables, and $|D| = n$, $x_i \in R^m$, and $y_i \in R$, the predicted value $\hat{y}_i$ for the $i^{th}$ sample, it can be represented by a model composed of $k$ decision trees, denoted as:

$$\hat{y}_i = \sum_{i=1}^{K} f_k(x_i), f_k \in F. \tag{6}$$

Here, $f_k$ represents the $k^{th}$ decision tree, and $F$ is a function space representing the collection of all decision trees.

Unlike the objective function of GBDT, XGBoost incorporates an additional regularization term on top of the original objective function to reduce overfitting and enhance generalization. The objective function is as follows:

$$L = \sum_{i=1}^{n} l(y_i, \hat{y}_i) + \sum_{k=1}^{K} \Omega(f_k). \tag{7}$$

Here, function $l$ can select different loss functions, while $\Omega(f_k)$ represents the penalty term for the $k^{th}$ tree. The specific formula is as follows:

$$\Omega(f_k) = \gamma T_k + \frac{1}{2} \lambda \sum_{j=1}^{T_k} w_{k,j}^2, \tag{8}$$

where $w_{k,j}^2$ represents the weight of the $j^{th}$ leaf in the $k^{th}$ tree, $T$ represents the number of leaf nodes, and $\gamma$ and $\lambda$ are parameters used to balance importance. By approximating the $L$ equation using a second-order Taylor series expansion, the following formula is obtained:

$$\begin{aligned}
L^{(t)} &= \sum_{i=1}^{n} l(y_i, \hat{y}_i) + \sum_{k=1}^{K} \Omega(f_k) \\
&= \sum_{i=1}^{n} l\left(y_i, \hat{y}_i^{(t-1)} + f_t(x_i)\right) + \sum_{k=1}^{K} \Omega(f_k) \\
&\approx \sum_{i=1}^{n} \left[ l\left(y_i, \hat{y}_i^{(t-1)} + g_i f_t(x_i) + \frac{1}{2} h_i f_t^2(x_i)\right) + \sum_{k=1}^{K} \Omega(f_k) \right].
\end{aligned} \tag{9}$$

Here, $g_i = \partial_{\hat{y}_i^{(t-1)}} l(y_i, \hat{y}_i)$ and $h_i = \partial^2_{\hat{y}_i^{(t-1)}} l(y_i, \hat{y}_i)$ respectively represent the first- and second-order approximations of the loss function $L$ with respect to $y_i^{(t-1)}$.

This model possesses the following advantages:

- The model supports parallel computation, resulting in higher computational efficiency.
- The algorithm supports column sampling, which reduces overfitting and enhances generalization ability while effectively reducing computational complexity.

- It incorporates a mechanism for handling missing values, enabling the model to automatically learn the splitting directions of tree nodes where there is missing data.
- Unlike GBDT, which solely employs first-order derivative information, XGBoost uses a second-order Taylor series, enabling the model to capture more granular data patterns and enhancing its accuracy.
- By incorporating L1 and L2 regularization into the loss function, the model's generalization ability is significantly improved.

The constructed OW-XGBoost model will be backtested to demonstrate its superiority over YL-XGBoost and MLC-XGBoost, as well as highlight its robustness.

## Model backtesting

The process of backtesting the model can be divided into two parts: stock selection and portfolio optimization. Given the current difficulties and limitations with short-selling stocks in the Chinese market, for present purposes, no consideration is given to shorting stocks that may decline. For the model's specific implementation process, we adopted a sliding window research method on the basis that it retains the time series characteristics of the data and better simulates the actual investment process. It should also be noted that ST stocks and stocks listed for less than 6 months were excluded from trading. The commission fee was set to 0.03% of the transaction value for buying and 0.03% of the transaction value plus 0.1% for stamp duty for selling, with a minimum commission deduction of 5 RMB per transaction.

### Selection of underlying asset

Considering the causal relationship between features and labels, the feature matrix was generated using market monthly data from the preceding 2 months on the first trading day of each month. To obtain the fusion label, the stock data from 1 month ago was combined with the optimal weight vector to form the training set for the training model. The monthly data from 1 month ago was then used as the prediction set to obtain the prediction results from the model and select stocks with predicted values above a set threshold for investment.

Furthermore, to enhance the machine learning model's performance, a grid search was carried out to adjust the hyperparameters of the model during the backtesting process. It should be highlighted that using the sliding window method to conduct the backtesting produces different optimal parameters for each rebalancing.

### Portfolio optimization

After obtaining the target stocks, the minimization of portfolio variance was used to determine the investment portfolio weights. Specifically, we utilized optimization algorithms to solve for the weights that minimize portfolio variance based on the historical returns and standard deviations of the selected assets over the preceding 250 trading days. The quantity of each stock to buy was then determined based on these weights. Unlike the equal-weighted investment portfolios commonly used in previous studies, this portfolio

approach takes into account the risk characteristics of different assets, thereby improving the investment portfolio's performance.

## RESULTS AND ANALYSIS

### Data source and preprocessing

#### Sample selection

By the end of 2023, the number of listed companies in China's A-share market had exceeded 5,000, with dozens of indices, including industry indices, size indices, and thematic indices. Among them, the CSI 300 Index's constituent companies are characterized by their larger scale, strong liquidity, and balanced industry coverage. As the total market value generally accounts for more than 60% of the total A-share market, it can accurately reflect the overall performance and trends of the Shanghai and Shenzhen stock markets. Moreover, the CSI 300 Index serves as an essential benchmark for index funds and derivative products, functioning as an important reference for evaluating investment strategy performance. On this basis, it plays a significant role in the entire financial market. The CSI 100 Index companies have capitalizations and higher market values and liquidity compared to those in the CSI 300 Index. Furthermore, the CSI 800 Index covers large, medium, and small capitalization companies, providing a more comprehensive industry layout. Among them, small and medium-sized enterprises have lower market value but greater growth potential.

As there might be differences in the stock price change patterns of companies of different scales, we selected the constituents of the CSI 300 Index as our main research subject. Additionally, the CSI 100 Index and CSI 800 Index constituents were used to test the robustness of OW-XGBoost.

The main research period was set for October 2022 to April 2023. In this period, the prevailing trend in the stock market was dominated by small- to moderate-amplitude fluctuations before later becoming relatively stable, which is aligned with most stock market conditions and holds research value. In addition, the stock market performance in the 2013–2015 period was more complex, such that it can be used to test the robustness of OW-XGBoost. Although the CSI 300 Index remained relatively stable overall from 2013 to mid-2014, from July 2014 to 2015, the Chinese A-share market experienced its last large-scale bull-to-bear transition (excluding during the pandemic), followed by a rebound at the end of the year.

#### Data preprocessing and factor selection

1) Data exclusion: Special treatment (ST) stocks with operational issues and newly listed stocks with high volatility were excluded.
2) Missing value handling: Any data with a high percentage of missing values for factors and labels was removed. Where the data featured a low percentage of missing values for factors, the missing values were filled in with the median.
3) Outlier treatment: The median absolute deviation (MAD) method was employed to handle the data. MAD is a statistical measure used to assess the degree of variation in a dataset. The specific calculation method is as follows:

**Table 1 Backtesting data for the optimal weighting selection.**

| Weight | Rate of return | Excess return | $\alpha$ | $\beta$ | Volatility | Sharpe ratio | Information ratio |
|---|---|---|---|---|---|---|---|
| [0.4, 0.4, 0.1, 0.1, 0] | 30.09% | 22.85% | 0.547 | 0.339 | 0.183 | 3.113 | 2.425 |
| [0.8, 0, 0, 0, 0.2] | 16.79% | 10.29% | 0.273 | 0.162 | 0.091 | 3.125 | 1.297 |
| [0.2, 0.3, 0.2, 0.2, 0.1] | 5.09% | −0.76% | 0.042 | 0.177 | 0.071 | 0.764 | −0.100 |
| [0.1, 0.3, 0.2, 0.4, 0] | 12.23% | 5.98% | 0.184 | 0.123 | 0.081 | 2.362 | 0.735 |
| [0.3, 0.2, 0.4, 0.1, 0] | 17.40% | 10.86% | 0.284 | 0.187 | 0.099 | 3.016 | 1.373 |
| [0.6, 0.2, 0, 0, 0.2] | 13.11% | 6.81% | 0.193 | 0.247 | 0.113 | 0.750 | 0.854 |
| [0.1, 0.4, 0.1, 0.2, 0.2] | 4.77% | −1.06% | 0.034 | 0.200 | 0.087 | 0.555 | −0.135 |
| [0.4, 0.5, 0.1, 0, 0] | 9.23% | 3.16% | 0.107 | 0.387 | 0.142 | 0.938 | 0.393 |
| [0.2, 0.1, 0.2, 0.1, 0.4] | 14.14% | 7.79% | 0.220 | 0.157 | 0.085 | 2.717 | 0.987 |

$$MAD = \mathrm{median}(|X_i - \mathrm{median}(X)|),$$
$$X'_i = \begin{cases} \mathrm{median}(X) + 5 \times MAD, \text{if } X_i > \mathrm{median}(X) + 5 \times MAD \\ \mathrm{median}(X) - 5 \times MAD, \text{if } X_i < \mathrm{median}(X) - 5 \times MAD \\ X_i, \text{if others} \end{cases} \tag{10}$$

4) Neutralization: The factor values were regressed against the industry average, with the residuals of the regression then taken as the processed factor values.

5) Standardization: The factor data was subjected to standardization processing. The standardization formula is as follows:

$$x_{scale} = \frac{x - \bar{x}}{\sigma_x}. \tag{11}$$

6) Feature selection: Initial feature selection was accomplished through variance filtering and the mutual information method. In total, 58 factors were chosen, encompassing quality, size, valuation, profitability, growth, liquidity, and technical aspects.

## Parameter selection

### Optimal weight selection

The grid search technique was employed to determine the optimal weights. Table 1 displays a selection of representative outcomes.

As can be seen from Table 1, under the weight distribution of [0.4, 0.4, 0.1, 0.1], OW-XGBoost significantly outperforms other weight distributions across multiple metrics. Specifically, it achieves a return rate of 30.09%, an excess return rate of 22.85%, an $\alpha$ value of 0.547, a Sharpe ratio of 3.113, and an information ratio of 2.425. These results exemplify OW-XGBoost's exceptional capacity to extract returns and excess returns.

### Trading frequency selection

To control trading costs and ensure the present research is comparable with others, the optimal trading frequency was selected based on monthly and weekly portfolio adjustments (Table 2).

**Table 2 Backtesting data for OW-XGBoost under different trading frequencies.**

| | Rate of return | Annualized return | Excess return | $\alpha$ | $\beta$ | Volatility | Sharpe ratio | Max drawdown |
|---|---|---|---|---|---|---|---|---|
| Monthly rebalancing | 30.09% | 61.05% | 22.85% | 0.547 | 0.339 | 0.183 | 3.113 | 5.9% |
| Weekly rebalancing | 18.57% | 36.15% | 11.97% | 0.302 | 0.287 | 0.146 | 2.202 | 6.13% |
| CSI 300 Index | 5.89% | 10.93% | | −0.811 | 1.237 | 0.169 | 0.143 | 8.78% |

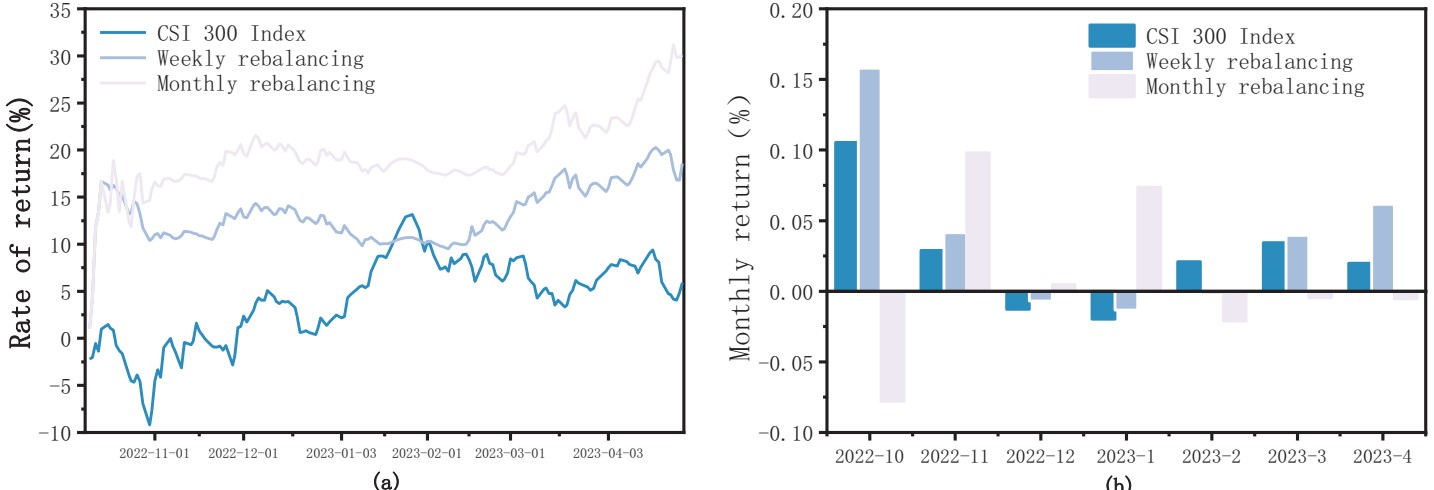

**Figure 1** (A) OW-XGBoost returns curve under different trading frequencies; (B) OW-XGBoost monthly returns under different trading frequencies.

As shown in Fig. 1, the return curve of monthly adjustments using OW-XGBoost remained consistently higher than that of the weekly adjustments. In terms of return metrics, monthly adjustments with OW-XGBoost significantly outperformed weekly adjustments for the following reasons:

1) A decreased trading frequency reduces tax and trading expenses.
2) The model incorporates a substantial number of fundamental factors, thus exhibiting a high value-investing attribute. Weekly adjustments cannot adequately capture the benefits of value regression, resulting in lower returns than monthly adjustments when using OW-XGBoost.
3) Compared to weekly adjustments, monthly adjustments are better at balancing short-term market fluctuations and evaluating and adjusting portfolios over a longer time span, thereby minimizing the impact of short-term fluctuations on returns.

Regarding risk metrics, the volatility of monthly adjustments with OW-XGBoost is slightly higher than that of weekly adjustments, though the maximum drawdown is slightly lower. Both exhibited robust risk-control capabilities, though there are minor differences in risk metrics.

**Table 3 Backtesting data for OW-XGBoost and YL-XGBoost.**

|  | Rate of return | Annualized return | Excess return | $\alpha$ | $\beta$ | Volatility | Sharpe ratio | Maximum drawdown |
|---|---|---|---|---|---|---|---|---|
| OW-XGBoost | 30.09% | 61.05% | 22.85% | 0.547 | 0.339 | 0.183 | 3.113 | 5.9% |
| YL-XGBoost | 10.71% | 20.23% | 4.55% | 0.131 | 0.458 | 0.190 | 0.853 | 8.68% |
| CSI 300 Index | 5.89% | 10.93% |  | −0.811 | 1.237 | 0.169 | 0.143 | 8.78% |

In conclusion, monthly adjustments with OW-XGBoost outperformed weekly adjustments in terms of return rate, excess return rate, and maximum drawdown, demonstrating superior overall market performance. On this basis, monthly adjustments were selected. Also, regardless of the trade frequency, the performance of OW-XGBoost significantly surpassed that of the CSI 300 Index, demonstrating and underscoring the efficacy of fusing labels.

## Comparison with YL-XBGoost

In order to compare the differences between OW-XGBoost, the CSI 300 Index, and YL-XGBoost, the market performance of the three models was analyzed. The method of minimizing portfolio variance was used to determine the investment weights of the targets for both OW-XGBoost and YL-XGBoost. Specific backtesting data can be found in Table 3.

It can be seen from Fig. 2 that the return trend of OW-XGBoost is typically higher than that of YL-XGBoost and the CSI 300 Index. Except for October 2022, the cumulative return rate of OW-XGBoost exhibited only minor fluctuations and drawdowns, broadly maintaining an upward trajectory. Where there is minimal volatility and maximum drawdown, it consistently achieves excess returns relative to the benchmark index. These results effectively demonstrated OW-XGBoost's ability to achieve high return rates while controlling for low volatility and a low maximum drawdown.

In terms of return rate, OW-XGBoost achieved a return rate of 30.09% during the backtesting period, with an annualized return of 61.05% (approximately five times the benchmark return). This performance is significantly higher than both YL-XGBoost and the CSI 300 Index, yielding an excess return rate of 22.85% during the backtesting period. Looking at the entire 7 months of backtesting, 4 months witnessed considerable return growth, with increases of more than 4%. The highest increase was in October, with 15.80%. Contrastingly, the 3 months with declining returns saw reductions of less than 1.5%, two of which experienced drops of less than 0.7%. It is noteworthy that in 3 of the 4 months when OW-XGBoost's returns increased, the CSI 300 Index experienced falling returns. This illustrates that OW-XGBoost remains able to effectively identify potential growth opportunities and achieve significant return growth in a bearish market. Meanwhile, YL-XGBoost's return fluctuations were strongly correlated with index movements, showing a greater vulnerability to market volatility.

Regarding volatility, YL-XGBoost showed the highest volatility, followed by OW-XGBoost. This indicates that YL-XGBoost's risk is further elevated, which imposes certain limitations. The overall volatility of OW-XGBoost is slightly higher than that of the CSI

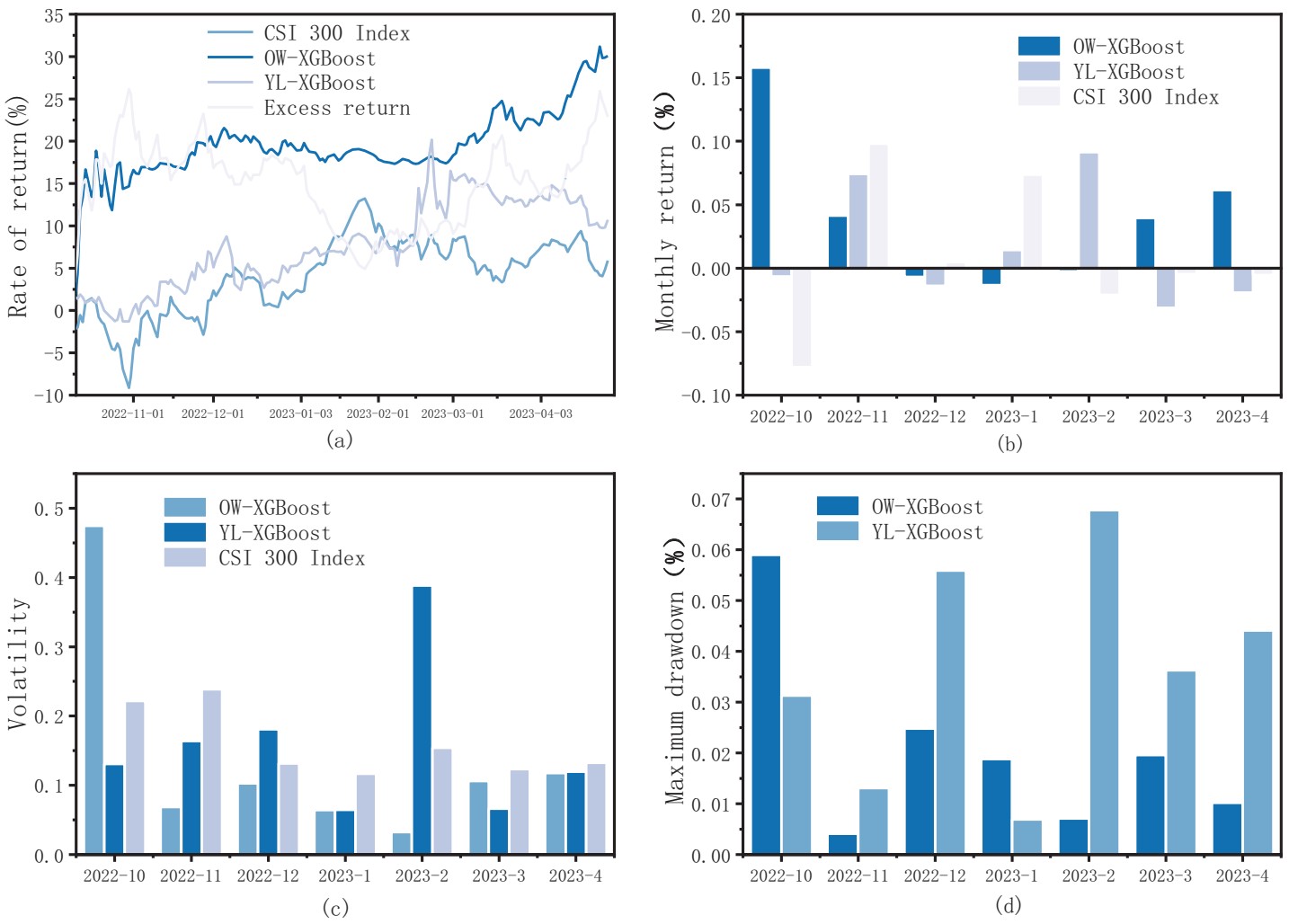

**Figure 2** (A) OW-XGBoost and YL-XGBoost returns curves; (B) OW-XGBoost and YL-XGBoost monthly returns; (C) OW-XGBoost and YL-XGBoost monthly volatility; and (D) OW-XGBoost and YL-XGBoost monthly maximum drawdown.

300 Index, though this can largely be attributed to the rapid return increase in October 2023. As returns and volatility are inherently interconnected, a consistently rising return curve can still generate relatively high volatility, even where high returns are achieved in the short term. This is the case with OW-XGBoost: when examining monthly volatility, OW-XGBoost's volatility is lower than that of the index for all months except October, revealing its effective ability to withstand market risks once fusing labels are introduced.

In terms of maximum drawdown, OW-XGBoost was significantly lower than both the CSI 300 Index and YL-XGBoost. Moreover, it demonstrated effective drawdown risk control in five of the monthly drawdowns, where it was notably lower than YL-XGBoost. An examination of OW-XGBoost's market performance reveals that its maximum drawdown, highest return, and highest volatility all occurred in October. This suggests that OW-XGBoost sacrifices some risk-control capacity in the pursuit of high returns.

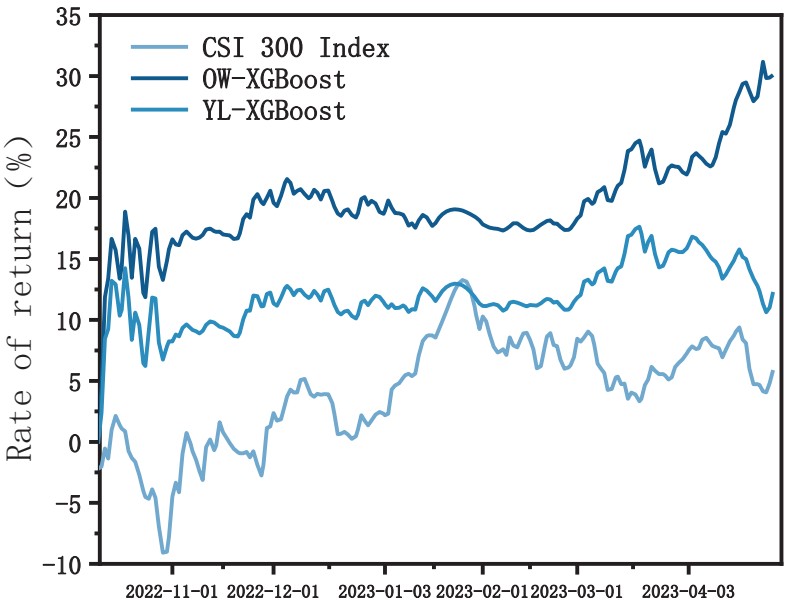

**Figure 3  OW-XGBoost and MLC-XGBoost returns curves.**

Furthermore, OW-XGBoost's beta value is 0.339, significantly lower than 1, reflecting its diminished susceptibility to market impact and increased resistance to market shocks. In comparison, YL-XGBoost's beta value is 0.458, indicating a higher vulnerability to market fluctuations. Meanwhile, OW-XGBoost's alpha is 0.131, reflecting a strong capacity to generate excess returns, and its Sharpe ratio stands at 3.113, indicating that for every unit of risk, 3.113 units of return can be achieved, which is an exceptional risk-reward ratio. Additionally, OW-XGBoost has a win rate and profit-loss ratio of 0.667 and 11.773, respectively; both values are significantly higher than those of YL-XGBoost. This suggests that OW-XGBoost has a greater capacity for risk control and profit-making compared to using the return rate alone as a benchmark.

## Comparison with MLC-XGBoost

In order to assess the superiority of OW-XGBoost over MLC-XGBoost, the backtesting results for both models were compared. The multi-label classification algorithm also used XGBoost, as shown in Fig. 3, to maintain comparability. After obtaining the classification results using the multi-classification algorithm, the score for each target was calculated based on the optimal weight. Ultimately, only the targets that achieved high scores surpassing the predetermined threshold were chosen to be included in the investment portfolio.

The overall trends exhibited by OW-XGBoost and MLC-XGBoost are somewhat similar, though the cumulative return curve of OW-XGBoost remained higher than that of MLC-XGBoost. Table 4 shows that both the return and excess return of OW-XGBoost are higher than those of MLC-XGBoost, though its volatility and maximum drawdown are

**Table 4 Backtesting data for OW-XGBoost and MLC-XGBoost.**

|  | Rate of return | Annualized return | Excess return | $\alpha$ | $\beta$ | Volatility | Sharpe ratio | Maximum drawdown |
|---|---|---|---|---|---|---|---|---|
| OW-XGBoost | 30.09% | 61.05% | 22.85% | 0.547 | 0.339 | 0.183 | 3.113 | 5.9% |
| YL-XGBoost | 12.30% | 23.39% | 6.05% | 0.162 | 0.454 | 0.173 | 1.124 | 7.03% |
| CSI 300 Index | 5.89% | 10.93% |  | −0.811 | 1.237 | 0.169 | 0.143 | 8.78% |

lower. These results demonstrate that the OW-XGBoost-based model significantly outperforms the MLC-XGBoost model in terms of earning returns and controlling risk.

For present purposes, the OW-XGBoost model offers the following advantages: First, it can comprehensively consider the interrelated returns and risk indicators in the market, meaning it is better able to capture the correlation patterns among different indexes, avoid the loss of related information, and provide a holistic assessment of the investment targets. Second, the model makes the multi-label problem easier to solve by turning it into a single-label classification problem based on optimal weights. This simplifies the model structure, lowers the risk of overfitting, improves generalization and stability, and makes the problem easier to solve. Third, utilizing fusing labels can better leverage the feature extraction capabilities of machine learning models, making the trained targets exhibit superior consistency and overall coordination, which in turn elevates the model's stability and risk resistance.

## Comparison with baseline models

To further analyze OW-XGBoost's performance in the stock market, five baseline models were picked for comparison with the XGBoost model: random forest (OW-RF), decision tree (OW-DT), support vector machine (OW-SVM), logistic regression (OW-LR), and ordinary least squares linear regression (OW-OLS). The results are shown in Fig. 4. Specific backtesting data can be found in Table 5.

OW-XGBoost's returns significantly surpassed those of the other models. The remaining five baselines exhibited remarkably high trend similarity, as can be seen in how the support vector machine, random forest, and decision tree baselines nearly overlap. These five baselines generally aligned with the market trend in the initial stage, though the fluctuation became less significant during the rising and falling phases, illustrating the efficacy of models using fusing labels in controlling risk. The cumulative return of these baselines gradually increased in the later stage, all of which surpassed the index return. The OLS regression model had the lowest return curve for most of the study period, with the lowest gain among the six models, just exceeding the CSI 300 Index (Fig. 4). Moreover, compared to other baselines, its return took around 20 days longer to turn from negative to positive.

The support vector machine model generated the highest profits, with a back-testing return of 17.35% and an annualized return of 33.62%. The two baselines with the lowest returns are OLS regression (13.44%) and logistic regression (13.21%). With the exception of OW-XGBoost in October, the monthly returns of other models fluctuated considerably

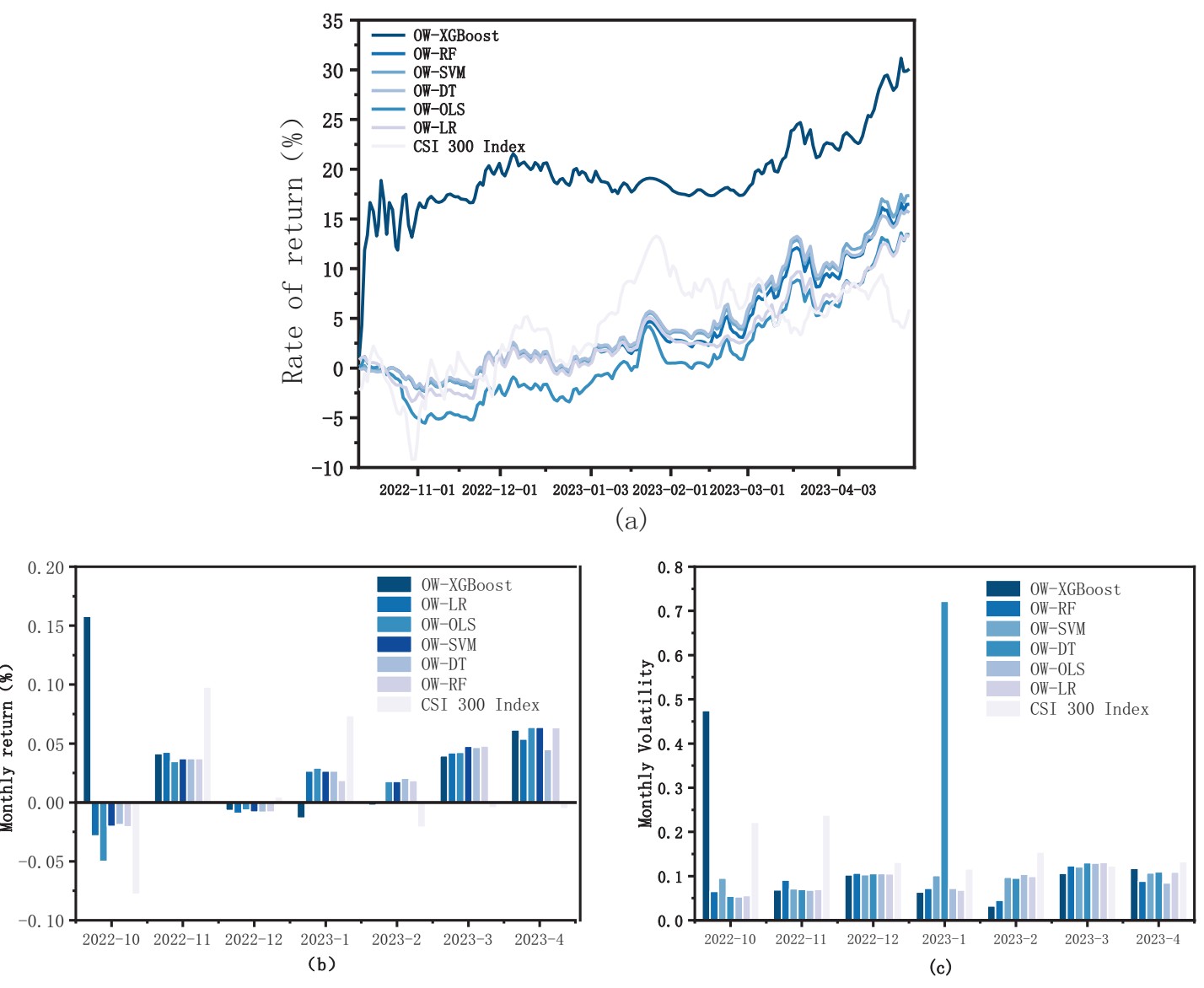

**Figure 4 Baseline models.** (A) Returns curves; (B) monthly returns; and (C) monthly volatility.

**Table 5 Backtesting data for baseline model.**

|  | Rate of return | Annualized return | Excess return | α | β | Volatility | Sharpe ratio | Maximum drawdown |
|---|---|---|---|---|---|---|---|---|
| OW-XGBoost | 30.09% | 61.05% | 22.85% | 0.547 | 0.339 | 0.183 | 3.113 | 5.9% |
| OW-RF | 16.52% | 31.91% | 10.03% | 0.266 | 0.195 | 0.099 | 2.833 | 3.01% |
| OW-DT | 15.63% | 30.09% | 9.19% | 0.249 | 0.169 | 0.094 | 2.784 | 2.99% |
| OW-SVM | 17.35% | 33.62% | 10.82% | 0.283 | 0.184 | 0.097 | 3.053 | 2.99% |
| OW-LR | 13.21% | 25.21% | 6.91% | 0.201 | 0.156 | 0.091 | 2.324 | 4.17% |
| OW-0LS | 13.44% | 25.67% | 7.13% | 0.204 | 0.188 | 0.103 | 2.111 | 6.62% |
| CSI 300 Index | 5.89% | 10.93% |  | −0.811 | 1.237 | 0.169 | 0.143 | 8.78% |

**Table 6 Backtesting data for OW-XGBoost after adjusting backtesting time.**

|  | Rate of return | Annualized return | Excess return | $\alpha$ | $\beta$ | Volatility | Sharpe ratio | Maximum drawdown |
|---|---|---|---|---|---|---|---|---|
| OW-XGBoost | 66.27% | 19.11% | 12.44% | 0.100 | 0.488 | 0.188 | 0.804 | 23.61% |
| CSI 300 Index | 47.88% | 10.93% |  | −2.134 | 1.008 | 0.283 | 0.435 | 62.93% |

less than the CSI 300 Index, demonstrating their capacity to remain profitable during market declines.

The overall volatility and maximum drawdown control abilities of the baselines did not vary significantly, and all were notably less than the CSI 300 index. Among the five baselines, the largest drawdown was OLS regression (6.62%), while the smallest was decision tree and support vector machine (2.99%). Most baselines maintained a maximum drawdown level of 3–4%, which is less than half of the CSI 300 Index's ultimate drawdown of 8.87%. In terms of overall volatility, the five baselines closely align, with OLS regression and logistic regression exhibiting the highest (0.103) and lowest (0.091) values, respectively. The remaining baselines' volatility fell within the 0.094–0.099 range, slightly exceeding half of the CSI 300 Index's volatility.

In summary, fusing labels based on optimal weights showed stable performance across various machine learning models, achieving superior returns compared to the market as well as lower market volatility and drawdown, thus demonstrating their effectiveness. It was discovered that nonlinear machine learning models did better than linear OLS regression and generalized linear logistic regression models in terms of return and risk control. This suggests that there are nonlinear relationships between factors and labels that linear models had trouble detecting. When it comes to linear models, generalized linear machine learning models do better than traditional linear OLS regression models. This is because machine learning models are better at finding linear relationships between factors. Finally, all models, even the traditional OLS regression, did better than the CSI 300 Index. This shows that combining labels based on optimal weights is a good way to show how return and risk work in the stock market, which in turn makes model investments perform better.

## ROBUSTNESS TEST

### Adjusting the backtesting period

To further investigate the robustness of OW-XGBoost, a longer duration characterized by a vibrant stock market, namely, 2013–2015, was selected to explore OW-XGBoost's performance in the context of complex market conditions. Specific backtesting data can be found in Table 6.

As can be seen in Fig. 5, from 2013 to mid-2014, OW-XGBoost was consistently higher than the Shanghai and Shenzhen 300 Index overall, with both less volatility and drawdown. During the bull market phase from July 2014 to June 2015, OW-XGBoost largely charted the same trend as the index. In the initial phase, OW-XGBoost's cumulative return remained above the index, though the gap gradually narrowed, and it was eventually

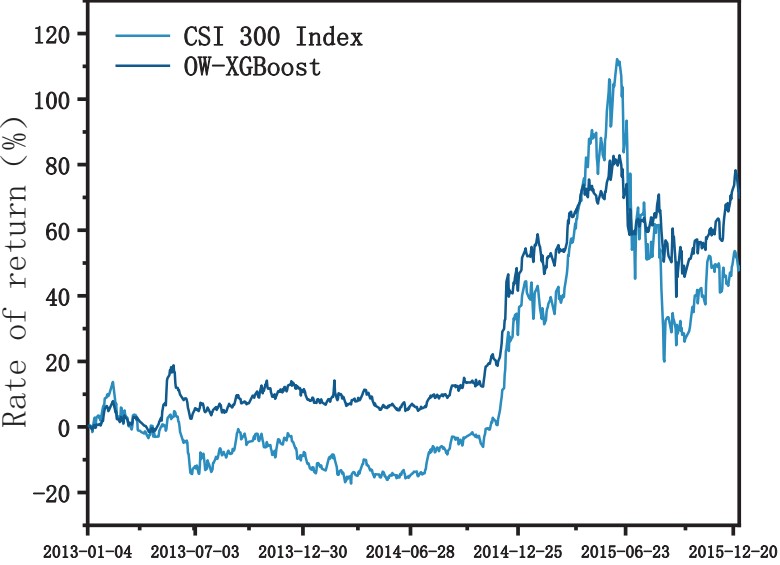

**Figure 5 OW-XGBoost returns curves after adjusting backtesting period.**

overtaken by the index in March 2015. During the stock price crash from June to September 2015, OW-XGBoost's drawdown was significantly less than the Shanghai and Shenzhen 300 Index, at around one-third of the maximum index drawdown. Later, during the bounce at the end of 2015, OW-XGBoost also showed superior performance. However, during the bull market, although OW-XGBoost's risk control performance significantly surpassed the market, the ultimate return was no better than that of the broader market. This could be due to the generally high stock price evaluation during the bull market: when OW-XGBoost needed to balance risk and return, its ability to unearth excessive returns in the stock selection process may have been negatively impacted.

In conclusion, based on OW-XGBoost's ability to achieve better return levels while controlling for volatility and a maximum drawdown smaller than the broader market in stable, turbulent, and drastic falling stages, it can be seen to possess a high level of robustness.

## Changing benchmark index

In order to assess the roburstness of OW-XGBoost across various stock pools, its effectiveness was additionally evaluated using stock pools consisting of constituent stocks from the CSI 100 and CSI 800 indices, as commonly done in literature. Specific backtesting data can be found in Tables 7 and 8, respectively.

### CSI 100 index

As shown in Fig. 6, the returns of OW-XGBoost consistently surpassed the index returns, with the gap between them rapidly widening early in the backtesting period. The backtest yield for OW-XGBoost stands at 30.96%, with an annualized yield of 63% and an excess yield rate of 24.63%. During the backtesting phase, the overall volatility of OW-XGBoost was slightly higher than that of the CSI 100 index; however, once October was excluded,

**Table 7  Backtesting data for OW-XGBoost with CSI 100 Index as stock pool.**

|  | Rate of return | Annualized return | Excess return | $\alpha$ | $\beta$ | Volatility | Sharpe ratio | Maximum drawdown |
|---|---|---|---|---|---|---|---|---|
| OW-XGBoost | 30.96% | 63.00% | 24.62% | 0.564 | 0.483 | 0.207 | 2.847 | 5.90% |
| CSI 100 Index | 5.09% | 9.41% |  | −1.061 | 1.360 | 0.181 | 0.117 | 9.95% |

**Table 8  Backtesting data for OW-XGBoost with CSI 800 Index as stock pool.**

|  | Rate of return | Annualized return | Excess return | $\alpha$ | $\beta$ | Volatility | Sharpe ratio | Maximum drawdown |
|---|---|---|---|---|---|---|---|---|
| OW-XGBoost | 13.98% | 26.76% | 6.81% | 0.214 | 0.156 | 0.091 | 2.509 | 2.79% |
| CSI 800 Index | 6.71% | 12.49% |  | −0.532 | 1.094 | 0.157 | 0.179 | 7.43% |

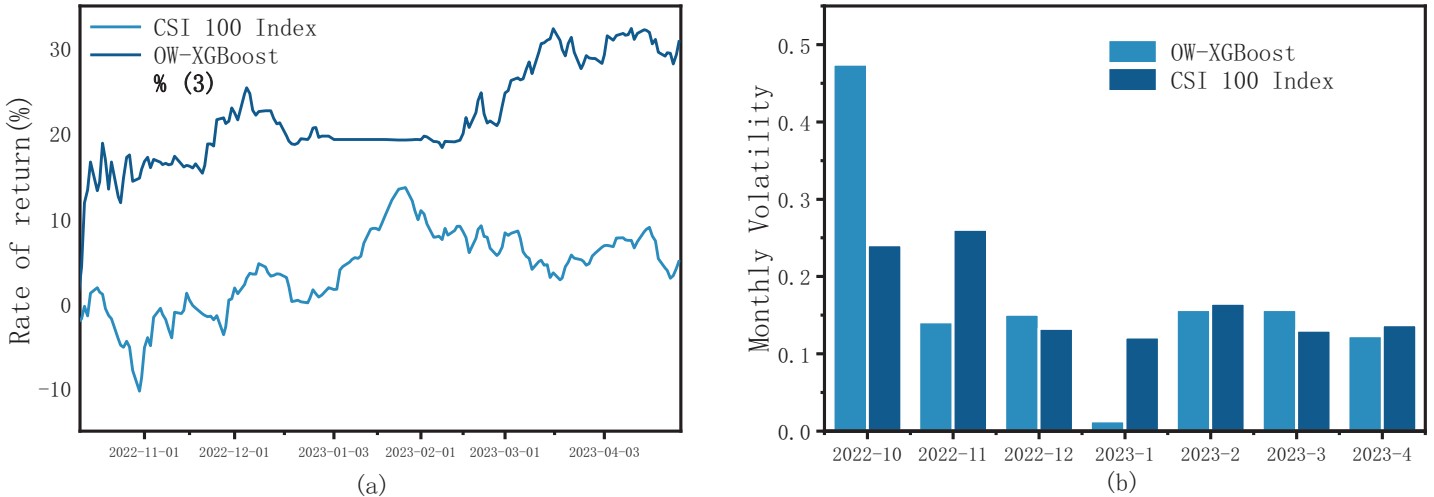

**Figure 6** (A) OW-XGBoost returns curves with CSI 100 index as stock pool; and (B) OW-XGBoost monthly returns with CSI 800 index as stock pool.

the overall volatility of OW-XGBoost fell below the index. In terms of monthly volatility, OW-XGBoost's volatility was lower than the index's volatility for 4 months. Meanwhile, the maximum drawdown for OW-XGBoost during the backtesting period was 5.9%, which was lower than the 10.08% maximum drawdown for the CSI 100 index.

### CSI 800 index

As shown in Fig. 7, the OW-XGBoost and CSI 800 index return curves show a significant degree of overlap. However, in comparison to the fluctuating index, OW-XGBoost maintained an overall upward trajectory. Additionally, it displayed a largely stable or improved performance throughout the three significant market fluctuations that occurred in October 2022, December 2022 to January 2023, and March to April 2023, without notably suffering from these fluctuations.

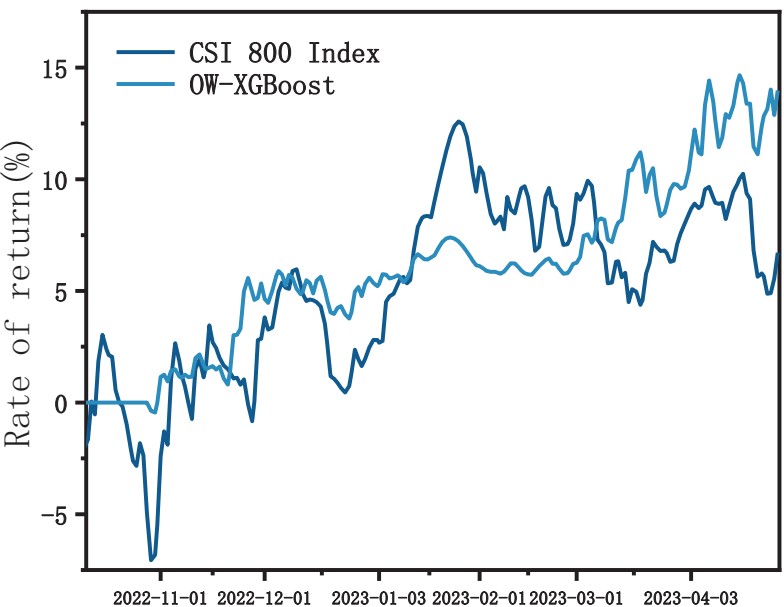

**Figure 7 OW-XGBoost returns curve with CSI 800 index as stock pool.**

During the backtesting period, OW-XGBoost's returns were approximately twice that of the index, with a volatility of 0.901, which is about three-fifths of the index. Furthermore, the maximum drawdown was only 2.79%, which is significantly lower than the index's maximum drawdown.

Comparing OW-XGBoost's performance in relation to the CSI 300, 100, and 800 indices, it can be found that OW-XGBoost performs better in backtesting when using high-market capitalization stocks as the stock pool. There are several reasons for this: first, studies have shown that in addition to the value of the company itself, small and medium-sized stocks in the Chinese stock market also possess a certain "shell value." However, neither fundamental nor technical factors are capable of reflecting this part of the value. As a result, OW-XGBoost struggles to accurately predict the ups and downs of small and medium-sized stock prices based on existing factors. Second, compared to large-cap companies, smaller-cap companies have more limited information disclosure, and the probability of irregularities in the disclosure process is higher, resulting in relatively lower information quality. Consequently, the characteristic factors that form on the basis of this information are also less likely to adequately predict trends. Third, large-cap stocks in the Chinese stock market have higher liquidity, resulting in relatively lower price fluctuations and less susceptibility to the daily limit system, which facilitates the OW-XGBoost fully realizing its potential.

OW-XGBoost has been tested in the past using stocks from the CSI 100 and CSI 800 indexes as stock pools. This shows that it is very good at finding extra returns and managing risks, and there is a lot of evidence to support this.

**Table 9 Backtesting data for OW-XGBoost under different training set durations.**

|  | Rate of return | Annualized return | Excess return | α | β | Volatility | Sharpe ratio | Maximum drawdown |
|---|---|---|---|---|---|---|---|---|
| 2 months | 17.95% | 34.85% | 11.38% | 0.296 | 0.174 | 0.105 | 2.939 | 3.00% |
| 3 months | 18.03% | 35.03% | 11.46% | 0.286 | 0.352 | 0.122 | 2.547 | 3.96% |
| 6 months | 16.19% | 31.24% | 9.73% | 0.244 | 0.406 | 0.148 | 1.836 | 8.93% |
| The CSI 300 Index | 5.89% | 10.93% |  | −0.811 | 1.237 | 0.169 | 0.143 | 8.78% |

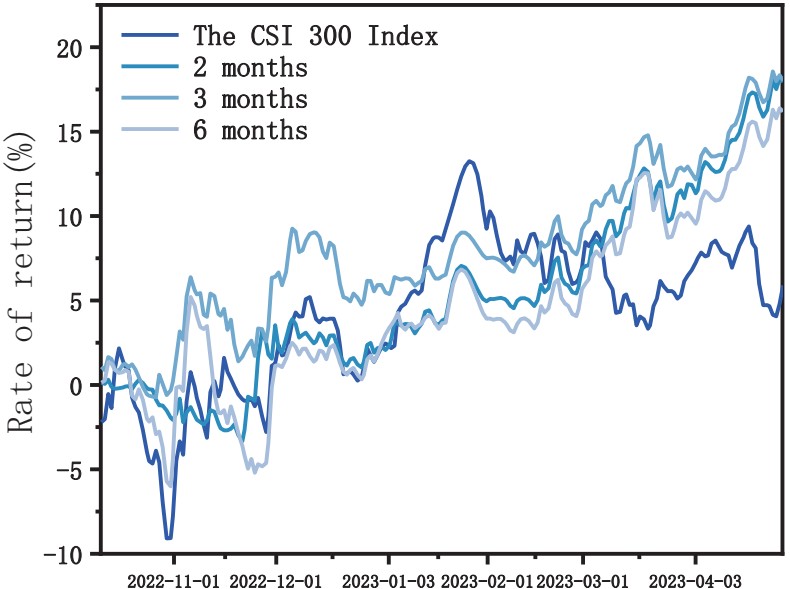

**Figure 8 OW-XGBoost returns curve under different training set durations.**

## Adjusting training set duration

To further examine the robustness of OW-XGBoost under different training set durations, model training was carried out using training sets with durations of 2, 3, and 6 months, respectively. Specific backtesting data can be found in Table 9.

As shown in Fig. 8, the trends of OW-XGBoost in the three scenarios are relatively similar, all achieving higher returns than the Shanghai and Shenzhen 300 Index and exhibiting lower volatility and maximum drawdown than the index. Hence, it can be concluded that OW-XGBoost can maintain excellent performance in terms of generating profits and controlling risks under different training set durations, indicating its robustness.

As the training set duration lengthens, OW-XGBoost's overall performance steadily declines for the following reasons: First, extending the length of the training set may cause OW-XGBoost to pay too much attention to noise or specific historical patterns in the historical data, thereby increasing complexity and overfitting while decreasing its generalization ability. Therefore, its performance in backtesting is relatively poor. Second, over time, the effectiveness of feature factors may change due to the influence of market

conditions and investor behavior. Longer training sets will give OW-XGBoost more early data, but they will not look into enough of the factors that work best in new market conditions or how these factors relate to each other. This means that OW-XGBoost does not work as well as it could.

## DISCUSSION AND CONCLUSIONS

By considering the portfolio's returns and risks comprehensively, a method based on OW-XGBoost is proposed, achieving outstanding results compared to YL-XGBoost, MLC-XGBoost, and five baselines. Specifically, OW-XGBoost can effectively control the impact of various risk and return indicators on the model by using fusing labels based on optimal weights. This alleviates the issues of label correlation and imbalance while improving the model's interpretability and generalization ability. In addition, OW-XGBoost can account for the volatility and instability of the stock market, demonstrating robust performance across different market conditions, stock pools, and training set durations. To be precise, the model performs optimally in mildly oscillating stock markets, in relation to stock pools consisting of high-market capitalization stocks, or with training set durations measured in months. Moreover, OW-XGBoost can provide a new perspective for quantitative research and the asset management industry, disrupting the inertia of selecting stocks solely based on maximizing returns and considering both risks and returns in this process. The concept of "fusing labels" was introduced to enrich the relevant theories for constructing portfolios that balance risks and returns. This approach also provides a feasible direction for implementing the national AI strategy in the financial field.

Though this study has made advances, it has also shown that OW-XGBoost still has some issues. While OW-XGBoost generally performs well in different types of stock market conditions, it still experienced significant drawdowns during the sharp decline in the A-share stock market in the second half of 2015. Future studies should incorporate trading instructions, such as profit-taking and stop-loss, and further explore OW-XGBoost and factor selection to enhance the model's ability to withstand systemic market risks.

### Funding
The author received no funding for this work.

### Competing Interests
The author declares that they have no competing interests.

### Author Contributions
- Jichen Liu conceived and designed the experiments, performed the experiments, analyzed the data, performed the computation work, prepared figures and/or tables, authored or reviewed drafts of the article, and approved the final draft.

## Data Availability

The Python code and data used in the experiments are available in the Supplemental Files.

## Supplemental Information

Supplemental information for this article can be found online at http://dx.doi.org/10.7717/peerj-cs.1931#supplemental-information.

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
