# Peer review of "Predicting Chinese stock market using XGBoost multi-objective optimization with optimal weighting"

_PeerJ Computer Science, doi:10.7717/peerj-cs.1931_

## Round 0.1 · original submission · Major Revisions

Please revise the manuscript to address the comments from the reviewers. Especially, please consider the suggestion from Reviewer 1 to "perform one more experiment to analyze the effectiveness of the size of the training set" and include more discussion and clarification about the motivation and findings of the study, as suggested by Reviewer 2.

Reviewer 1 ·

Basic reporting

The manuscript was well-written with clear and unambiguous technical language. The introduction and background sections provide sufficient context for the study, and the relevant prior literature has been appropriately referenced throughout the manuscript. The figures are of high quality, in vector format, and are nicely displayed in the manuscript. They are relevant to the content of the manuscript and appropriately described and labeled.

The code was distributed as a supplemental document, making it easily accessible and reproducible. The manuscript is "self-contained" and covers all important results, and it clearly matches the scope of the journal.

Experimental design

The research question is defined clearly, and the study makes a valuable contribution to filling a knowledge gap. The methods have some novelties in terms of using the fused labels, and they were described clearly in the manuscript. The study should be reproducible, as the code was provided. It is beneficial to the literature as the software is reproducible and shows good results.

However, there are some unclear points in the method that the author should clarify. They are listed below.
- The author mentioned using the Median Absolute Deviation method for outlier treatment. However, it is unclear how this method works.
- Why XGBoost was chosen for modeling? What are the advantages of this algorithm?

One more thing: although the author has done some experiments to show the robustness of the proposed method, I suggest the author perform one more experiment to analyze the effectiveness of the size of the training set. This can be done by using different durations in the time-series data when defining the training set.

Validity of the findings

An additional experiment as suggest above could be conducted to further demonstrate the statistical soundness of the results. The conclusions were appropriately stated, connected to the original question investigated, and limited to those supported by the results.

Additional comments

A revision is needed for this manuscript.

Reviewer 2 ·

Basic reporting

The manuscript looks well-structured and written in a formal English manner. The literature study is adequate and provides a thorough overview of the topic. The tables and figures are well-designed and represent the research findings in a clear manner. Despite these advantages, the manuscript still requires considerable work before it can be considered ready for publication. Specifically, there are several points listed below that need to be clarified.

Experimental design

The method is described clearly in the manuscript, and it is technically sound. The study should be reproducible, as the code was provided. However, the manuscript should be better if the authors provide more information about the following:
- Motivation for using fused labels when developing their models.
- Motivation for using the selected risk and return indicators.

Validity of the findings

The findings appear to be valid and can be valuable for academics from other domains who are interested in stock prediction or other time-series prediction concerns. The dataset and code are uploaded with the study, making it easier for others to replicate the findings and expand on the research.
However, the study should be better if the authors address the following:
- About the results in Figure 1, what could be the reasons that the monthly adjustments with OW-XGBoost significantly outperformed the weekly adjustment for return metrics?
- About the results in Table 4, what could be the reasons that OW-XGBoost outperformed MLC-XGBoost?

Additional comments

- The introduction of the study can be improved, and I think more relevant studies can be included in this part.
- What are possible formulas for the penalty term in Equation (7)?

---

## Round 0.2 · Minor Revisions

Please revise the manuscript in accordance with the reviewers' comments, especially to discuss the limitations of the method. Also, please ensure that any errors in the list of references are corrected.

Reviewer 1 ·

Basic reporting

No comment

Experimental design

No comment

Validity of the findings

No comment

Additional comments

The author has addressed all of my comments, resulting in significant improvements to the manuscript. While the current version can be considered for publication, I recommend that the author incorporate a discussion regarding the limitations of the method to enhance the comprehensiveness of the study.

Reviewer 2 ·

Basic reporting

The author has added more details and conducted additional experiments to improve the quality of the manuscript. The manuscript is now well-structured and understandable to readers. I recommend that this work be considered for publication.

Experimental design

I have no more comments.

Validity of the findings

The work provides sufficient results to evaluate the validity of the findings. I have no more comments.

Additional comments

Some terms and keywords in the references need to be capitalized:
+ Li, X. and Sun, Y. (2020): "rbf" should be "RBF".
+ Liu, B., Li, M., Ji, Z., Li, H., and Luo, J. (2024): "ai" should be "AI".
+ Xu, X. and Ghosh, M. (2015): "lasso" should be "LASSO".

---

## Round 0.3 · accepted · Accept

The author has addressed all of the reviewers' comments. As the comments were minor, there is no need to resend the manuscript to the reviewers. The manuscript is ready for publication.